# Organic Matter in the Asteroid Ryugu: What We Know So Far

**DOI:** 10.3390/life13071448

**Published:** 2023-06-26

**Authors:** Christian Potiszil, Masahiro Yamanaka, Chie Sakaguchi, Tsutomu Ota, Hiroshi Kitagawa, Tak Kunihiro, Ryoji Tanaka, Katsura Kobayashi, Eizo Nakamura

**Affiliations:** The Pheasant Memorial Laboratory for Geochemistry and Cosmochemistry, Institute for Planetary Materials, Okayama University, Yamada 827, Misasa, Tottori 682-0193, Japan

**Keywords:** Ryugu, Hayabusa2, organic matter, origin of life, prebiotic chemistry, asteroid, comet, astrobiology, amino acid, sample return

## Abstract

**Simple Summary:**

Organic matter is an integral part of all living things on our planet, but abiotic (not relating to life) organic matter can be found throughout the Solar System within meteorites, asteroids, comets and even on the surfaces of other planets such as Mars. Therefore, one of the most profound questions for humanity concerns how life originated on Earth. In order to answer this, we must first understand how organic matter is formed and altered throughout our solar system and the universe in general. To better understand the origin and evolution of organic matter, the Japan Aerospace Exploration Agency (JAXA) sent the Hayabusa2 spacecraft to collect and return material from the near-earth asteroid Ryugu. Here, a review concerning the organic material within the returned Ryugu samples will be presented. The review will inform the reader about the Hayabusa2 mission, the nature of the organic matter analyzed and the various interpretations concerning the analytical findings. The review puts the findings and individual interpretations in the context of the current theories surrounding the formation and evolution of Ryugu and highlights the issues that remain to be solved.

**Abstract:**

The Hayabusa2 mission was tasked with returning samples from the C-complex asteroid Ryugu (1999 JU3), in order to shed light on the formation, evolution and composition of such asteroids. One of the main science objectives was to understand whether such bodies could have supplied the organic matter required for the origin of life on Earth. Here, a review of the studies concerning the organic matter within the Ryugu samples is presented. This review will inform the reader about the Hayabusa2 mission, the nature of the organic matter analyzed and the various interpretations concerning the analytical findings including those concerning the origin and evolution of organic matter from Ryugu. Finally, the review puts the findings and individual interpretations in the context of the current theories surrounding the formation and evolution of Ryugu. Overall, the summary provided here will help to inform those operating in a wide range of interdisciplinary fields, including planetary science, astrobiology, the origin of life and astronomy, about the most recent developments concerning the organic matter in the Ryugu return samples and their relevance to understanding our solar system and beyond. The review also outlines the issues that still remain to be solved and highlights potential areas for future work.

## 1. Introduction

### 1.1. The Hayabusa2 Mission

The Hayabusa2 mission was the successor of the Hayabusa mission [1], which returned samples from the asteroid Itokawa in 2010. The Hayabusa mission found that Itokawa was composed of ordinary chondrite material and had experienced numerous impacts by hyper velocity particles [2]. In contrast, the Hayabusa2 mission was tasked with returning samples from an asteroid that was thought to be more primitive in nature, potentially resembling carbonaceous chondrite-like material [3]. As such, a major scientific goal of the mission was to investigate the organic matter present in the asteroid Ryugu, in terms of its origin, evolution and potential to have supplied the early Earth with organic molecules relevant to the origin of life.

The Hayabusa2 spacecraft was launched in December 2014 by a H-IIA rocket and after 3.5 years reached its home position ~20 km above Ryugu [4]. Much was deduced from remote sensing of Ryugu, including its geomorphology, albedo, bulk density and the thermal properties and composition of the material at its surface [4,5,6,7]. Accordingly, Ryugu was found to have a “spinning top shape”, with a pronounced equatorial ridge that yielded an equatorial radius of 502 m. The estimated bulk density, based on the volume and gravity measurements of Ryugu, was 1190 kg m^−3^.

In addition to the remote sensing, several touchdown sites were selected and two were picked for the collection of the samples to be returned to Earth (Figure 1) [8,9]. The first touchdown site (TD1) was situated at 4.30° N and 206.47° E of the equatorial region on the surface of Ryugu and the touchdown maneuver was carried out on the 21st of February 2019 [8]. The sampling mechanism involved the firing of a tantalum bullet into the surface on touchdown [3], in order to break up the material and allow collection of the fragments into chamber A of the Hayabusa2 spacecraft [8]. The onboard camera (CAM-H) recorded the whole maneuver and revealed that the sub-surface material was darker than that of the surface and that the displaced material had a bright and dark side, with the bright side being a similar albedo to the surface of Ryugu [10].

The second touchdown site (TD2) was selected for the creation of an artificial impact crater, which would allow access to subsurface materials that should be less affected by space weathering [9]. The artificial impact crater was produced on 5 April 2019 via the collision of a 2 kg copper projectile, traveling at 2 km s^−1^, with the surface of Ryugu [12]. The impact produced a semi-inverted conical crater with an apparent diameter of 14.5 m and a depth of 1.7 m, from the original surface. The crater was located at 7.90° N and 313.30° E of the equatorial region on the surface of Ryugu and the touchdown maneuver was completed on 11 July 2019 [9]. The TD2 samples were collected 22 m from the artificial impact crater into chamber C and it was estimated that subsurface material, deposited as ejecta, should have been collected in a ratio of 1:3 compared to the surface material [13]. The calculation was based on modeling the ejection of material from the crater and the maximum depth that the ejecta could have come from was estimated to be ~1.3 m. As such, the TD2 samples should record both surface and subsurface materials.

After completing its remote sensing and sample collection operations, the Hayabusa2 spacecraft returned the Ryugu samples to Earth on the 6th of December 2020, in a hermetically sealed container [14]. Subsequently, it was found that 5.4 g of the sample had been recovered with 3.2 g in chamber A (from TD1) and 2.0 g from chamber C (from TD2) and the rest is found within the sealed container outside of either chamber.

### 1.2. The Origin and Evolution of Extraterestrial Organic Matter

#### 1.2.1. The Interstellar Medium

Extraterrestrial organic matter (OM), such as that found within the Ryugu samples and carbonaceous chondrites, is composed of two main fractions. The first fraction is soluble in common lab solvents and not bound to a macromolecular structure and is thus termed either soluble organic matter (SOM) or free organic matter (FOM) [15,16]. The FOM/SOM fraction accounts for around 30% of the OM in carbonaceous chondrites [15]. The second fraction is insoluble in common lab solvents and consists of a macromolecular structure comprising both aliphatic and aromatic moieties [15,16]. Thus, the second fraction is termed as either insoluble organic matter (IOM) or macromolecular organic matter (MOM) and accounts for the majority (~70%) of OM in carbonaceous chondrites.

The OM in extraterrestrial objects is thought to originate in a number of distinct environments, ranging from the interstellar medium (ISM) and the protosolar nebula (PSN) to the progenitor planetesimals of asteroids/comets and their daughter meteorites (Figure 2) [16,17,18,19]. The organic-forming elements (organic elements hereafter), with the exception of H and D, are synthesized in stars and emitted into the ISM via the formation of planetary nebulae, in the case of asymptotic-giant branch (AGB) stars, and through supernovae for main sequence stars 10× more massive than our sun [20]. Some organic molecules formed in the atmospheres of stars, such as fullerenes and polyaromatic hydrocarbons (PAHs), may also be distributed into the ISM during the formation of a planetary nebula [21,22]. Meanwhile, all H and D in our universe were formed in the big bang and have been used as fuel in stars ever since.

Once in the ISM, the organic elements can be present in their atomic form in diffuse atomic clouds or when the clouds are denser in their molecular forms, as is the case for molecular clouds [23]. Molecular clouds can be further split into diffuse molecular clouds, where many of the organic elements heavier than H will be in their atomic forms, and dense molecular clouds where the organic elements will be present in their molecular forms [23]. The chemistry in diffuse molecular clouds is dominated by intense UV interactions and only large organic molecules, such as PAHs, and particularly resistant small molecules can survive [24].

Meanwhile, dense molecular clouds offer protection from UV irradiation within their interiors, which prevents the destruction of organic molecules formed within them [23]. Organic synthesis in dense molecular clouds can proceed through three main processes: gas-phase chemistry, gas-grain chemistry and irradiation-driven chemistry involving ice [23,25]. In the case of gas-phase chemistry, reactions occur when several molecules collide, and these can be between either neutral molecules (neutral–neutral) or a neutral molecule and an ion (ion–molecule) [25]. Neutral–neutral collisions rarely result in a reaction, due to the large activation energies required. Whereas ion-molecule reactions do not require activation energy and are thus more likely to result in a reaction [23]. Currently, around 256 different organic species have been identified in the ISM, with many of these being in the gas phase [26].

Furthermore, the ability of dense molecular clouds to protect their interiors from UV irradiation, allows their gasses to condense to form ice as mantles on dust grains through gas-grain chemistry [27,28,29]. The ice has been found to consist of a number of volatiles in addition to water including CO, CO_2_, HCO, H_2_CO, CH_3_OH, CH_4_ and NH_3_ [23,30]. As such, the effect of forming ices from gaseous molecules leads to the formation of simple organic molecules, including HCO, H_2_CO, CH_3_OH and CH_4_, as well as inorganic molecules that can go on to form OM upon further processing. The buildup of ice on dust grains also creates a matrix that stabilizes certain ions and radicals and allows for solid-phase reactions to occur [23].

Once the ice mantles have formed, subsequent photo-processing generates ions and radicals via cosmic rays and UV photons, produced by their interaction with H_2_ [23,31]. The radicals and ions can then interact in the ice to produce larger organic molecules or if separated by large distances can combine on melting of the ice, once it has been accreted into a parent body shortly after the formation of the Solar System. Indeed, on recovery of even simple irradiated ices (e.g., H_2_O:CH_3_OH:NH_3_:CO = 100:50:1:1) at room temperature, hundreds to thousands of organic molecules are produced [32]. Accordingly, many studies have investigated the potential for different ices to yield biologically important products. To date, a large variety of such compounds have been demonstrated to form during ISM ice analogue experiments, including amino acids [33,34] and sugars [35,36]. While nitrogen heterocycles, such as nucleobases, have not been observed in the gas phase in the ISM, they have been produced successfully in ISM ice experiments [37,38,39]. While it was previously found that no genetically relevant nucleobases could be formed without the addition of pyrimidine and purine [23,40], nucleobases were produced in a more recent study from irradiation of simple ices deposited from simple gas mixtures [39]. It was concluded that the lower temperature allowed for the presence of CO ice which may have affected the available reaction pathways, leading to the production of all the canonical nucleobases found in life. Therefore, it is most likely possible for all the building blocks of life, required to make genetic material and proteins, to form in the ISM.

#### 1.2.2. The Protosolar Nebula and Protostellar Disk

When a molecular cloud collapses to form a PSN and eventually a protostellar disk (PSD), the conditions will vary dramatically throughout the disk, with temperatures close to the protostar being hot enough to vaporize rocks, but the conditions at the middle of the PSN and further out along the mid plane being as low as 10 K [23,41]. As such, gas phase and gas-grain chemistry, similar to that described above for dense molecular clouds, can occur in the outer portion of a PSN/PSD [41,42]. However, due to the denser environment, almost all the photochemistry is confined to the disk surface above and below the midplane and the inner edge of the disk closest to the protostar. Additionally, unlike molecular clouds the PSD is expected to be turbulent [23,41]. When turbulence moves a dust grain with an ice mantle to the disk surface it will be exposed to UV and X-ray irradiation from the protostar, but this will be at much higher doses than for dense molecular clouds [43]. The higher radiation doses and thus temperatures will cause the ice to sublimate and leave behind radicals and ions which recombine to form a residue. The residue upon further irradiation has then been demonstrated from simulation experiments to undergo hydrogen loss and aromatization, which creates a material similar, in terms of its X-ray absorption near-edge structure (XANES) spectra, to the MOM/IOM found in carbonaceous chondrite meteorites [44].

While gas phase and gas-grain chemistry reminiscent of the ISM may be a driving force for OM formation in the outer PSN/PSD, a different style of these processes may also be operating within the photosphere throughout the disk. Plasma discharge experiments, referred to as the Nebulotron experiments, investigated the synthesis of OM in the high energy environments of the PSD photosphere [45,46,47,48,49]. During the interaction of UV and X-rays from a protostar with the surface of its PSN/PSD, a hot ionized gas is formed. Such a phenomena is simulated by inducing a plasma discharge into a gas mixture, consisting of CO, N_2_ and H_2_ [49]. At this point, the ionized gas reacts to form small organic molecules, which copolymerize to give larger OM that can coagulate to form solid particles [47]. If the solid OM sticks onto a dust grain surface, it can be transported by the turbulence in the PSD to a cooler region where it can survive [49]. Nevertheless, if the organic solid particles were further processed by X-rays and UV irradiation, similar effects to those hypothesized for photo-processed ice mantle residues can occur. Furthermore, the Nebulotron experiments have also produced a number of biologically important molecules within their soluble fractions and pyrolysates, including nucleobases, sugars and amino acids [49].

#### 1.2.3. Planetesimals

When the PSN/PSD enters into the protoplanetary disk phase, streaming instabilities or vortices cause dust, ice and any associated OM to coagulate to form cm-sized pebbles [50,51,52]. The pebbles can then form larger bodies called planetesimals via either gravitational collapse [50,53] or hierarchical agglomeration [54,55]. Planetesimals are thought to heat up due to the decay of short-lived radio nuclides, such as ^26^Al, included in their rocky dust components [20]. As a result, the ice melts producing aqueous fluids rich in organic and volatile components, which react with the minerals comprising the dust to form phyllosilicates and other secondary phases, including sulfides, metal oxides and carbonates [56,57,58]. During this period of aqueous alteration, a large array of different organic reactions can take place, potentially leading to the production of both the FOM/SOM and MOM/IOM found in carbonaceous chondrite meteorites [16,18,59,60].

The FOM/SOM component of carbonaceous chondrites records many biologically important compounds that could have formed during aqueous alteration [16]. Multiple aqueous phase reaction pathways can lead to the formation of amino acids from simple organic precursors, with Strecker synthesis yielding α-amino acids [61,62] and Michael’s addition of ammonia, with subsequent hydrolysis, producing β-amino acids [62,63]. Another potential route to amino acids could be the breakdown of the irradiated ice analogue component hexamethylenetetramine (HMT), which was shown to produce amino acids upon hydrothermal processing [64]. It may have also been possible for nucleobases to form during planetesimal aqueous alteration [65], either via HCN polymerization [66,67] or via the reaction of cyanoacetylene with cyanate in a slightly alkaline solution [68]. Similarly, sugars have been suggested to form during aqueous alteration via Formose reactions [69,70], which could have also been involved in the formation of amino acid precursors. As such, aqueous alteration, like ISM processes, could have led to the formation of all the building blocks of life required to produce proteins and genetic material.

The MOM/IOM component of carbonaceous chondrites has also been suggested to form during aqueous alteration, with Formose-type reactions, among others, being demonstrated as possible candidates [59,60]. Indeed, certain attributes of the MOM/IOM were recreated during experiments, including the formation of nano-globules, Fourier transform infrared (FTIR) response (e.g., CH_2_/CH_3_ and C=O/C=C), X-ray absorption near edge structure (XANES) spectra, ^13^C nuclear magnetic resonance (NMR) spectra and ^13^C isotopic composition [60,71]. Therefore, both MOM/IOM and FOM/SOM may have been formed on the planetesimal progenitor of carbonaceous chondrites and do not necessarily have to form solely in the ISM or PSN/PSD.

#### 1.2.4. Isotopic Signatures

The N and H isotope composition among Solar System materials varies greatly [72], with the δ^15^N and δD values for the Gas Giant Jupiter representing some of the lowest values in the solar system (δ^15^N = −374 ± 82‰ and δD = −833 ± 45‰) [73,74]. Interestingly, the values for Jupiter are similar to those proposed for the PSN (δ^15^N = −382 ± 8‰ and δD = −865 ± 32‰) obtained from solar wind values [75,76]. Meanwhile, some of the highest δ^15^N and δD values among solar system objects are recorded for carbonaceous chondrites (bulk δ^15^N = −23.7 to 359.7 and δD = −225.8 to 763.1) [77,78] and comets (δ^15^N = −10.8 to 2022.6 and δD = −120.5 to 641,000.0) [79,80]. While the δ^15^N and δD values of comets go up to much higher values than the bulk values from carbonaceous chondrites, there is significant overlap of the values between these two types of extraterrestrial objects at the lower end of the range for comets. To complicate the picture further, Wild2 cometary silicates from the Stardust mission record carbonaceous chondrite-like bulk δ^15^N values and also contain ^15^N-rich hotspots similar to those observed in carbonaceous chondrites [81,82]. 

Furthermore, there are far less measurements for comet N and H isotope compositions than for carbonaceous chondrites and many were undertaken using spectroscopy for specific species within cometary coma. As such, measurements for comet N and H isotopes are often associated with large uncertainties [72]. While the comet 67P represents an exception, having been studied in detail [80], it still only represents one comet. Therefore, it is difficult at present to know what the bulk N and H isotope compositions of an average comet would be or if major variations exist between comets for their ice and silicate phases. Nevertheless, it is possible to examine how the above isotopic enrichments observed in carbonaceous chondrites and comets may have arisen.

As mentioned above, the δ^15^N and δD values for PSN gases were likely very low [75,76]. As such, isotopic fractionation in the ISM, PSN or progenitor planetesimals of carbonaceous chondrites and comets is required to explain the δ^15^N and δD values of their enriched components. For D, enrichment can occur through many different processes in the ISM, including gas-phase ion-molecule reactions, gas-grain chemistry and photolytic chemistry [23,83]. In particular, gas-grain chemistry is thought to be very effective at enriching D and has been suggested to result in D/H ratios as high as 0.1 (δD = 641,000) [84], which is similar to the maximum value observed for cometary species [79,80]. Whereas, photolytic chemistry is capable of enriching organic molecules, such as polyaromatic hydrocarbons (PAHs), in D, but also pass along enrichments found within ices, which formed through gas-phase and gas-grain chemistry, to the new organic products [23,85].

For N, many of the above processes were previously proposed to result in meaningful isotopic fractionation and thus ^15^N enrichment [23]. However, for gas-phase chemistry, similar exchange reactions that were capable of enriching D, would not be able to proceed at the low temperatures required to produce large ^15^N enrichments. This is because there are barriers preventing the exchange reaction, as a result of the requirement for a pair of electrons to be broken and then reformed, which is not the case for D^+^ or H^+^ [23,86]. 

Similarly, for gas-grain chemistry, the enrichment of ^15^N may be limited. A large fractionation in H isotopes is possible because the mass difference between D and H is very large and this means the mobility of H in the ice is much greater than for D [23]. Accordingly, H will be more likely to interact with other H atoms, than D and form H_2_, which can escape as a gas. For N, the difference in the mass of its isotopes is small and so the difference in mobility between them is small. Additionally, the abundance of N is much lower than for H, so the initial reactions of N atoms will likely be with H [23].

For photolytic chemistry involving ices, initial enrichments in ^15^N can be passed along [23,40], but enrichment of organic molecules in the gas phase, as happens with PAHs for D, is likely, not possible. Instead, another photolytic process, N_2_ self-shielding (photodissociation) is thought to be responsible [72,87,88,89,90]. In this scenario, the outer regions of a molecular cloud or the regions of the PSN closest to the Proto-Sun are bombarded by UV radiation and the photons capable of dissociating a ^14^N-^14^N molecule are adsorbed within that region, generating gas-phase ions and molecules as a result. Interestingly, because the number of ^14^N-^15^N molecules is lower, the photons capable of breaking the bond within these molecules are able to penetrate further and thus enrich ^15^N in any ices or OM formed within the interior facing edge of the exterior of molecular clouds or the outer portion of the PSN [89]. As a result, the gas phase is depleted in ^15^N. In agreement, an experimental study found that huge enrichments of ^15^N were achieved during photodissociation of N_2_ [90] and these were found to be more than capable of explaining the enrichments recorded by components within both carbonaceous chondrites and comets [72,77,78,79,80]. However, molecular clouds are much larger than the PSN, which means that their interiors are not exposed to significant amounts of UV photons. This is likely why the exterior envelopes of dense molecular cores (dense molecular clouds) seemingly record higher ^14^N/^15^N ratios (^15^N depletion) in the gas-phase overall than their interiors [87,88].

Upon formation of the Solar System, comets and carbonaceous chondrite progenitor planetesimals would accrete the ^15^N and D enriched ice and dust. The presence of short-lived radionuclides leads to the heating and melting of the ice and the release of any OM contained within them [16,56]. At this stage, any new OM produced will inherit the isotopic enrichments. Nevertheless, it is also possible for further isotopic enrichments to occur during the formation of the OM. Indeed, it has been shown that carbon isotopes can be fractionated during polymerization of FOM/SOM to MOM/IOM, via a Formose type reaction [60]. As such, it may be possible for other isotopes to be fractionated as well during aqueous alteration on the planetesimal progenitors of carbonaceous chondrites and comets.

Finally, another mechanism that has been shown to produce isotopic enrichments in OM directly, is that of plasma discharge reactions [46,48,49]. At the surface of the PSN, UV and X-rays from the Proto-Sun will induce a plasma discharge, which is capable of generating OM that can condense onto the surface of dust grains [49]. The dust grains can then be transported to cooler regions of the PSN, where they can survive and be accreted by carbonaceous chondrites or comets. Experiments that have tried to recreate these conditions were capable of producing MOM/IOM enriched in D, with D-hotspots similar to those found in carbonaceous chondrites [46]. However, no evidence of ^15^N enrichment was found and thus N2 self-shielding remains the main process capable of generating enrichment of ^15^N in carbonaceous chondrites and comets. 

## 2. Ryugu (1999 JU3): Remote Sensing Observations and Predictions

The asteroid 1999 JU3, which would later be named Ryugu, was classified as a Cg-type asteroid in 2001, based on its strong UV absorption feature shortward of 0.55 um and its flat to slightly reddish slope longward of 0.55 um [91]. Cg-type asteroids are part of the C-complex of asteroids, which were suggested to be “primitive” in nature and potentially the parent bodies for carbonaceous chondrites [92,93]. The linking of carbonaceous chondrites and C-complex asteroids relates to several interpretations concerning features in the near infrared spectra of C-complex asteroids. The features were interpreted as arising from secondary alteration minerals, including goethite, hematite, jarosite and phyllosilicates, that are the products of aqueous alteration and which are found in carbonaceous chondrites [92,93].

Many carbonaceous chondrite meteorites contain significant amounts of OM [16], which made 1999 JU3 of interest to potential sample return missions that would like to investigate the origin and evolution of OM in our solar system. Additionally, 1999 JU3 represented a particularly accessible asteroid for a mission that would like to rendezvous with a C-complex asteroid [94]. Accordingly, 1999 JU3 was selected as the target of the Hayabusa2 mission [95].

In the time between the selection of the Hayabusa2 mission target and the rendezvous of the Hayabusa2 space craft with Ryugu, several ground-based investigations were undertaken to try and predict what the asteroid may be composed of. The first of these studies suggested that Ryugu was most similar to a heated CM2 chondrite composition, based on comparisons of the reflectance spectra of Ryugu, obtained from the Very Large Telescope in Chile to that obtained from two heated Murchison (CM2) samples (heated to 900 °C and 1000 °C) [96]. In contrast, a subsequent study concluded that Ryugu was actually more similar to an unheated specimen of the Mighei (CM2) meteorite [97].

As the Hayabusa2 spacecraft approached Ryugu, it was able to record spectra of the asteroid’s surface. As a result of the remote sensing investigation, the asteroid was reclassified to a Cb-type asteroid, which is similar to Cg-type asteroids but has a flat or linear slope and lacks the strong UV absorption feature of the Cg-type asteroids. Furthermore, the composition of Ryugu was estimated to be most similar to a moderately dehydrated carbonaceous chondrite, due to the absence of a ubiquitous 0.7 um absorption band (associated with Fe-rich serpentine, a hydrated mineral) and one of the lowest albedos of any solar system object [5].

Accordingly, a scenario was envisioned in which the asteroid Ryugu was formed through the catastrophic collision of a much larger asteroid, and which was subsequently gravitationally reaccumulated to form a rubble pile asteroid with a spinning top shape. Moreover, the dehydration was suggested to have occurred from internal heating, likely due to the progenitor planetesimal of Ryugu forming early in the history of the Solar system, when the short-lived radionuclide ^26^Al was still very abundant. In this scenario, OM should have also been altered by the heating and bear similarities to that of heated carbonaceous chondrites.

Interestingly, the study also noted that interplanetary dust particles (IDPs) have a similarly low albedo, but preferred heated carbonaceous chondrites, due to their lack of the 0.7 um absorption band [5]. However, Cb-type asteroids were previously linked to IDPs and comet-like icy asteroids [98,99] and so the connection between these extraterrestrial objects and Ryugu permitted further evaluation. Using previous ground-based and Hayabusa2 spacecraft remote sensing data and reflectance spectra from irradiated OM and carbonaceous chondrites, a series of mass balance equations were solved to predict the OM content of Ryugu [10]. The results suggested an incredibly high OM content for Ryugu of between 14.6 and 59.3 volume % (vol.%). Such a finding was backed up by video footage from the Hayabusa2 space craft, which showed displaced material from the first touchdown being lighter on the surface facing away from the asteroid and darker on that pointing towards the interior of Ryugu. The only explanation for a carbonaceous chondrite-like material becoming brighter upon irradiation was that OM had been converted to graphite. If Ryugu had such a high OM content, then it would appear to be IDP-like in nature. As IDPs had been previously linked to comets or described as comet-like [99,100,101], a cometary origin for Ryugu was suggested as an alternative to the catastrophic collision and gravitational reaccumulation model [10].

## 3. Organic Matter in the Ryugu Return Samples

### 3.1. Free/Soluable Organic Matter (FOM/SOM)

#### 3.1.1. Amino Acids

Amino acids were included in the first analytical study to report on the OM inventory of Ryugu [17]. The results indicated the presence of both proteinogenic (protein forming) and non-proteinogenic amino acids within both a Ryugu particle and a sample of the Orgueil (CI1) chondrite, confirming that Ryugu contained indigenous amino acids similar to those in the carbonaceous chondrites present in Earth collections. Furthermore, the study showed that the intensity of the amino acids within Orgueil was higher than in Ryugu and suggested this may be due to some terrestrial contamination effect for Orgueil, which had been present on the Earth for more than 150 years [102]. Nevertheless, the study provided no chirality (optical isomer) information and did not quantify the amino acid abundances of the Ryugu or Orgueil sample. Such information would be incredibly useful because L-amino acids are primarily utilized in the proteins of all living organisms on Earth and thus measurements of amino acid chirality can help studies to assess whether any terrestrial contamination is contributing to their results.

Subsequently, another study investigated the amino acids in Ryugu, including their abundances and their chirality [103]. Two different techniques were utilized to determine the amino acid abundances and perform the chiral separations, with a targeted analysis utilizing 3-dimensional high-performance liquid chromatography-fluorescence detection (3D-HPLC-FD) and a non-targeted analysis using liquid chromatography-fluorescence detection and high-resolution mass spectrometry (LC-FD/HRMS). The study yielded amino acid abundances that were significantly different between the two techniques (Figure 3), with a number of amino acids, such as glycine, D-alanine and D- and L-α-aminobutyric acid, being much lower for the LC-FD/HRMS technique than for the 3D-HPLC-FD technique. The authors attributed the lower glycine to the use of multiple vacuum drying steps in the LC-FD/HRMS analysis, which could have aided in the loss of volatile organic compounds that could have generated further glycine, during the alkaline conditions used during sample preparation for the 3D-HPLC-FD technique. 

However, a previous study [104], using a similar method to the LC-FD/HRMS technique, was able to detect very large abundances of glycine within an extract of the Orgueil meteorite (Figure 3). While the methods differ in the number of vacuum drying steps, with the Orgueil meteorite investigation using less, both methods used a vacuum drying step prior to any acid hydrolysis, which should have been enough to lose the most volatile organic compounds. Instead, the LC-FD/HRMS technique did not use a desalting step or any alkaline reagents, which may explain the discrepancy between it and the Orgueil investigation.

Meanwhile, the 3D-HPLC-FD technique utilized no initial vacuum drying step and only a very slight alkaline pH of 8. Nevertheless, the 3D-HPLC-FD technique gave a Ryugu glycine abundance that was lower than that reported in the Orgueil study. Previously, another study using essentially the same 3D-HPLC-FD technique reported abundance values for the Murchison meteorite [105]. Similarly, the same technique used by the Orgueil study [104] was applied to the Murchison meteorite. When comparing these values, they are largely the same, albeit with a slightly higher level of terrestrial contamination present for the 3D-HPLC-FD study, indicated by the higher L-enantiomer excesses (for alanine and valine). This result is interesting because it questions the importance of the vacuum drying steps and the volatile precursors in determining the glycine abundance and instead may point to the absence of a desalting step as the main issue, in the LC-FD/HRMS analysis utilized for the Ryugu amino acid abundance determination.

In a separate investigation of the amino acids within the Hayabusa2 return samples [106], both the hydrolyzed and unhydrolyzed fractions for two different samples were reported. The sample preparation technique applied was the same as that used for the LC-FD/HRMS analysis in the study mentioned above [103]. The results indicated that the unhydrolyzed fractions contained much higher abundances of amino acids than the hydrolyzed fractions, which is uncommon. The reason for using acid hydrolysis is to convert amino acid precursors to amino acids or liberate bound amino acids and thus improve the chance of detecting trace levels of amino acids in small sample sizes [107]. Accordingly, except in rare cases, such as for the achondritic ureilite Elephant Moraine (EET) 83,309 [108], hydrolyzed fractions usually yield higher abundances of amino acids compared to unhydrolyzed fractions.

The strange result for the Ryugu hydrolyzed fractions was interpreted to potentially arise due to either the degrading effect of small Fe-rich phases in the extracts or as a result of cations interfering in the derivatization of the amino acids. The authors favored the Fe-rich phases scenario because they did not believe the cations would be sufficient within the hydrolyzed fractions to cause a problem. However, it is curious that Orgueil contains largely the same modal mineralogy as Ryugu samples, yet previous analyses of the hydrolyzed fractions of this meteorite [104], using a very similar methodology, did not report lower amino acids within the hydrolyzed compared to unhydrolyzed fractions. Instead, the only major difference in the methodologies was the lack of any desalting step in that used for the Ryugu analysis. Therefore, it would seem likely that interference from cations in the derivatization for the hydrolyzed fractions was the main cause. Indeed, it is possible that some cation-containing phases, transferred with the hot water extracts, were dissolved to yield a greater contribution of cations during the acid hydrolysis step than when no acid hydrolysis was undertaken. As such, the amino acid abundance data for the Ryugu sample analyses that used the LC-FD/HRMS technique should be interpreted with caution and the unhydrolyzed data from the second study [106] used over the hydrolyzed data. Meanwhile, for the 3D-HPLC-FD technique [103], the hydrolyzed data is likely comparable to the Orgueil hydrolyzed data reported previously [104] and implies that there may be some differences in the amino acid abundances between the two extraterrestrial samples.

Furthermore, both the aforementioned studies provided invaluable chiral information, which indicates that the amino acids in Ryugu are likely mostly racemic. The exception is several protein-bearing amino acids and a non-protein-bearing amino acid [106]. However, the enantiomeric excesses were largely observed within the hydrolyzed fractions and as such, it is not clear if there was contamination introduced during this step or if any indigenous bound or amino acid precursor material contained an enantiomeric excess. The lack of more widespread L-enantiomeric excesses questions the idea that extraterrestrial environments, such as those with circularly polarized light [109], may give rise to L-excesses and explain the L-amino acids observed in the proteins of living organisms on the Earth. Nevertheless, the 3D-HPLC-FD technique only targeted a very small number of amino acids and thus it is necessary for future work to constrain the issue concerning the LC-FD/HRMS technique or develop a new analytical technique to determine amino acid chirality in the Ryugu samples more reliably.

Finally, leading on from the qualitative investigation of amino acids in Ryugu reported previously [17], the qualitative data from this study and data from a new Ryugu sample were investigated to quantify the amino acid abundances in Ryugu (Figure 3) [18]. In the study, no effort to constrain the chirality of Ryugu amino acids was undertaken. Instead, a focus on determining the amino acid abundances reliably and in the context of the mineralogy and petrology of the samples was prioritized. The findings of the study indicated that two Ryugu particles from different touch-down sites contained different abundances of amino acids, but this was not likely as a result of differing levels of irradiation exposure, as had been suggested for other organic compounds [17,110].

**Figure 3 life-13-01448-f003:**
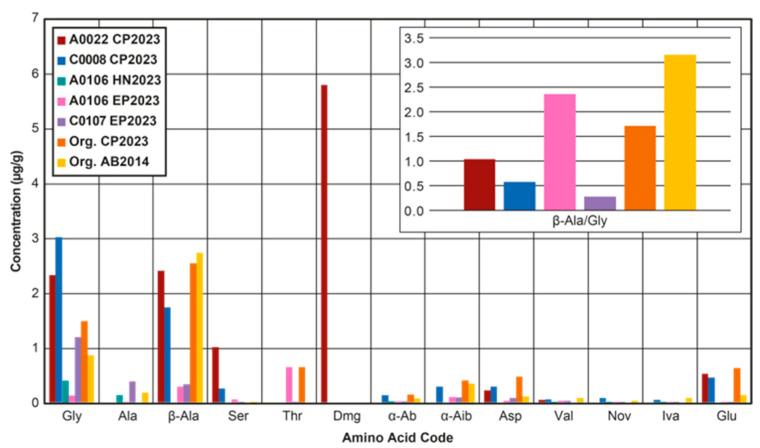
A bar chart demonstrating the amino acid abundances in parts per a million (µg/g) for the three studies that have reported such data for Ryugu. Also included are values for the Orgueil carbonaceous chondrite (CI1). To the top right-hand side of the diagram is another bar chart showing the β-alanine/glycine ratio for two of the studies, which targeted both of these amino acids. Gly = glycine, Ala = alanine, β-ala = β-alanine, Ser = serine, Thr = threonine, Dmg = dimethylglycine, α-Ab = α-aminobutyric acid, α-Aib = α-aminoisobutyric acid, Asp = aspartic acid, Val = valine, Nov = norvaline, Iva = isovaline, Glu = glutaric acid. CP2023 = [18], HN2023 = [103], EP2023 = [106], and AB2014 = [104].

In contrast, the modal mineralogy and geochemistry of the two particles was significantly different, with the particle A0022 (TD1) having greater abundances of secondary alteration phases (i.e., carbonate, magnetite and Fe-sulfides) compared to C0008 (TD2) and evidence for mobilization of its trace elements over the mm scale, which was not seen for C0008. The differences were interpreted as arising from a greater accretion of ice (including CO or CO_2_ ice) for A0022, compared to C0008 (Figure 4). The greater ice content led to higher levels of aqueous alteration and the generation of higher abundances of secondary phases for A0022 than C0008, including carbonate that could have formed from the accreted CO or CO_2_. Indeed, the β-alanine/glycine ratio, commonly used to interpret the level of aqueous alteration for carbonaceous chondrites, was found to be higher for A0022 than C0008, indicating it had likely experienced more extensive aqueous alteration (Figure 3).

As such, A0022 may have accreted greater levels of irradiated ice products, such as glycine, or glycine precursors, which could have gone on to react during the aqueous alteration to form another amino acid N, N-dimethylglycine (DMG). The amino acid DMG forms through the Eschweiler–Clark reaction and involves the reaction of glycine with formaldehyde and formic acid [111]. Both formaldehyde and formic acid were observed in the comet 67P [112] and thus should have been accreted by Ryugu. Accordingly, DMG was found to be the most abundant amino acid in A0022, with DMG being below detection limit in C0008, while glycine was lower in A0022 compared to C0008. Therefore, glycine may have been initially much higher in A0022 than C0008 but was used up during the reaction to form DMG. The results of the study indicate that the style of aqueous alteration was likely heterogenous throughout Ryugu and highlighted the importance of aqueous alteration in the determination of the final amino acid abundances of extraterrestrial objects, including those that likely seeded the early prebiotic Earth with OM.

**Figure 4 life-13-01448-f004:**
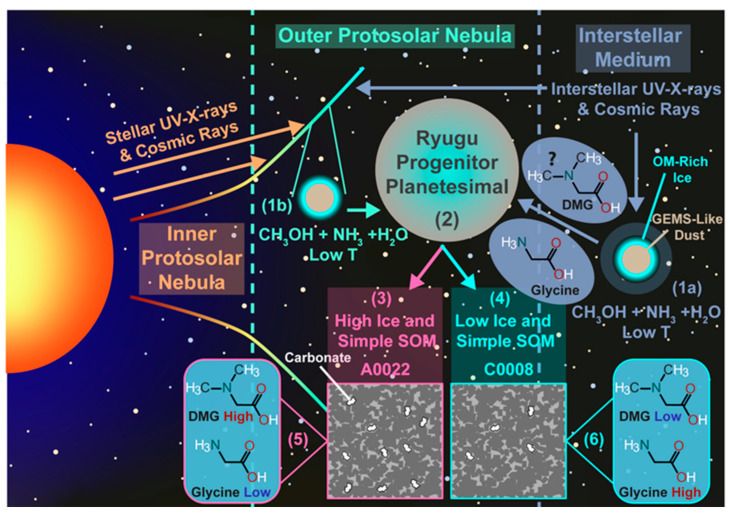
An overview of the processes affecting the amino acids, other organic matter and minerals within the Ryugu particles A0022 and C0008. The figure was reproduced from a previous study [18], without further modification. (1) OM can form in the interstellar medium (ISM) within molecular clouds prior to the formation of the Solar System (1a) or within the protosolar nebular (PSN) or protoplanetary disk (PPD) at low temperatures (Low T) (1b). (2) Amino acids and organic-rich ice formed prior to or during the accretion of the Ryugu progenitor planetesimal are accreted into the body, along with glass with embedded metal and sulfides (GEMS) and silicate minerals or amorphous silicates. Short-lived radionuclides, e.g., ^26^Al, decay and generate heat, which melts organic-bearing ice that leads to aqueous alteration and the alteration/generation of organic matter and minerals. (3) larger quantities of CO_2_-rich ice and potentially glycine is accreted into the region that later becomes A0022, compared to C0008 (4). (5) the glycine accreted by the Ryugu progenitor planetesimal, or which formed through Strecker synthesis during aqueous alteration, react with formaldehyde and formic acid in the water-rich fluids to generate dimethylglycine (DMG) at higher abundances in A0022 than in C0008 (6).

#### 3.1.2. Amines

Aliphatic amines were observed in Ryugu samples and demonstrated a decreasing abundance with increasing chain length, which is consistent with the trend observed for these compounds in other carbonaceous chondrites [103,106]. The detectable amines for both studies consisted of methylamine, ethylamine, isopropylamine and propylamine, in order of decreasing abundance. The higher abundance of the branched isopropylamine compared to propylamine was interpreted as potentially arising due to either a radical reaction or as a result of thermal processing, where the branched chained amines are more resistant to thermal degradation [103]. In terms of their formation, the amines may have been synthesized from the breakdown of amino acids [113], but the lack of isotopic data makes this hard to prove [106]. Furthermore, the comparison to other carbonaceous chondrites was hampered by the fact that most aqueously altered carbonaceous chondrites yield > 10 aliphatic amines and Ryugu only had 4 [106]. Additionally, even Orgueil records an additional aliphatic amine in the form of butylamine [114] and suggests some subtle differences in the OM between these mineralogically similar extraterrestrial specimens.

#### 3.1.3. Nitrogen Heterocycles

Nitrogen heterocycles (N-heterocycles) were reported previously for carbonaceous chondrites and were thus expected to be present within Ryugu [115,116,117,118,119,120]. Accordingly, several studies confirmed their presence within the Ryugu return samples [17,103,110]. The first study to report on N-heterocycles used desorption electrospray ionization-orbitrap-mass spectrometry (DESI-OT-MS) to generate ion intensity maps for Ryugu particles [17]. The study was able to calculate the chemical formulas of the ions detected and compare these to those observed for a previous ultrahigh performance liquid chromatography-orbitrap-mass spectrometry (UHPLC-OT-MS) [117] and DESI-OT-MS [119] study. The results indicated that Ryugu contained a large array of N-heterocyclic compounds that formed homologue series and appeared more similar to Murchison than Orgueil, in terms of the types identified. Orgueil contained a very limited number of N-heterocycles in comparison to the Ryugu particles and recorded only one homologue series, which was also observed for the Ryugu particles. Furthermore, the particle from TD1 contained an overall lower number of compounds within the homologue series detected, compared to the TD2 particle and also demonstrated a more constrained spatial distribution and overall lower intensity. The results were interpreted as arising due to differences in the irradiation exposure that the Ryugu particles had experienced. The TD1 particle may have been sampled from closer to the surface of Ryugu and thus exposed to higher levels of solar and galactic cosmic rays than the TD2 particle. Therefore, higher levels of irradiation for the TD1 particle may have led to the preferential destruction of the N-heterocycles in this particle, compared to the TD2 particle.

A similar conclusion was drawn concerning the results of another study which investigated the nucleobase and other N-heterocycle contents of Ryugu return samples [110]. The study investigated the nucleobase abundances within Ryugu extracts and found that the TD1 particle had lower abundances of Uracil compared to the TD2 particle. Nevertheless, due to the requirement for the extracts to also be analyzed for amino acids, an acid hydrolysis step was included. The inclusion of the acid hydrolysis step made the sample preparation different from that carried out in a previous analysis of carbonaceous chondrite nucleobases [118]. To interpret the potential differences introduced, the authors also analyzed the Orgueil (CI1) meteorite with and without the acid hydrolysis step. The results indicated that the uracil content increased by a factor of 1.5 after acid hydrolysis and so the hydrolyzed Ryugu abundances are expected to be similarly affected by acid hydrolysis. When taking this into account, Ryugu would have a much lower nucleobase abundance, in terms of both its free and bound/precursor fractions, compared to Orgueil. 

The values for Ryugu are actually much more similar to the unhydrolyzed values for CM2 chondrites and indicate that Ryugu in fact has a uracil content less than or similar to CM2 chondrites. No other nucleobases, apart from some uracil structural isomers and alkylated uracil homologues, were detected and this may relate to the acid hydrolysis step and/or the very low concentrations/absence of these nucleobases in Ryugu. Comparatively, CM2 chondrites were found to contain all the canonical nucleobases found in DNA/RNA [118]. However, it remains unclear whether CI chondrites contain a similarly diverse suite of nucleobases, as the results for these molecules were not reported for the Orgueil sample analyzed alongside Ryugu [110]. Such information would be incredibly useful to understand the effects of the acid hydrolysis step on the determination of other canonical nucleobases found in DNA/RNA on Earth and thus their possibility to be within the Ryugu samples.

In a further study, N-heterocyclic homologue series were reported for a Ryugu particle using UHPLC-OT-MS, nano-liquid chromatography-OT-MS (nanoLC-OT-MS) and DES-OT-MS [103]. The results demonstrated, similarly to those reported previously [17], that the Ryugu samples contained a large array of N-heterocycle compounds forming a variety of homologue series. While the study reported relative abundance distributions for the homologues, these should be interpreted with some caution because ion intensity will differ with increasing alkyl chain length, due to ionization efficiency-related effects [121]. Therefore, the distributions reported are not strictly relative abundance, but actually relative intensity. Nevertheless, it is clear from the study that the normalized ion intensity plotted against carbon number indicates a difference in the profiles between Ryugu and Murchison. Furthermore, the presence of homologues identified in CR chondrites within Ryugu may suggest differences in either the accreted OM or the parent body conditions (e.g., redox conditions or water/rock ratios) between Ryugu and Murchison [103].

#### 3.1.4. Other Compounds

Monocarboxylic acids (MCAs) are among the most abundant FOM/SOM observed in carbonaceous chondrites [122,123,124]. Despite this, only formic acid and acetic acid were detected above the detection limit in Ryugu [103]. The lack of other higher molecular weight MCAs might relate to the fact that formic acid and acetic acid are by far the most abundant MCAs in carbonaceous chondrites and only a small fraction (~50 µL) of the hot water extract was used to look for them. Moreover, the hot water extract for Ryugu utilized only ~13 mg of sample, while previous studies of carbonaceous chondrites used between ~80 mg and >20 g of the sample [122,123,124]. Nevertheless, formic acid and acetic acid were detected at high abundances, similar to those in Orgueil (CI1) and Ivuna (CI1) [103] and this indicates that Ryugu samples are likely less structurally diverse for MCAs compared to less aqueously altered carbonaceous chondrites. The low structural diversity was interpreted as a result of heightened aqueous alteration, which is known to lead to the destruction of MCAs.

Both alkylbenzenes and polyaromatic hydrocarbons (PAHs) were detected in Ryugu solvent extracts [103]. Interestingly, the four ring PAHs fluoranthene and pyrene were the most abundant and fluoranthene was much less abundant than pyrene. In Murchison, the aforementioned PAHs are present in equal amounts [125], but it has been proposed for Ivuna that pyrene may be separated from fluoranthene via geochromatography, relating to the different solubilities of the two isomers in fluids during aqueous alteration [126]. As such, it was interpreted that the differences in the abundances of pyrene and fluoranthene could have arisen during aqueous alteration of the progenitor planetesimal of Ryugu [103]. Indeed, geochromatography was also proposed to explain the distribution of N-heterocyclic compounds on the surface of Murchison samples [115,119].

### 3.2. Macromolecular/Insoluable Organic Matter (MOM/IOM)

#### 3.2.1. Micrometer Scale Observations

Micrometer scale observations for Ryugu MOM/IOM were first undertaken using scanning electron microscopy (SEM) with energy dispersive X-ray spectroscopy (EDS), Raman spectroscopy and Fourier transform infrared (FTIR) spectroscopy [17] (Figure 5). The SEM-EDS results indicated that MOM/IOM occurred as µm-sized carbonaceous rich areas, referred to as carbonaceous nodules, but also as distinct aggregates of OM with sharp boundaries, which were termed micro-OM.

Meanwhile, Raman spectroscopy, applied to 16 Ryugu particles, indicated that MOM/IOM was distributed ubiquitously throughout the matrix and revealed a slight difference between the particles in terms of their G-band FWHM and peak center positions. The G-band arises from the in-plane stretching of pairs of carbon sp^2^ atoms [127] and is particularly strong for aromatic-rich organic materials [128]. In Ryugu and carbonaceous chondrites the OM is composed of amorphous carbon, indicated by the nature of the G-band and the presence of another band, termed the D-band [17,128]. The D-band arises only in aromatic materials and relates to defects in the crystal lattice of the material, such as those introduced by sp^3^ orbitals or heteroatoms [127]. When an organic material is irradiated its structure is altered and the full width at half maximum (FWHM) and peak center of the G-band is changed. The differences observed for Ryugu were consistent with a more irradiated end member for the TD1 particles A0022 and A0073 and a less irradiated end member for the TD2 particles C0082 and C0079 [17]. The differences in irradiation were interpreted as arising due to disparate irradiation fluxes between more deeply buried TD2 particles and particles from TD1 and TD2 that were sourced from close to the surface. The abundance estimates of Ne from cosmic rays for each particle also agreed with the evidence provided by Raman spectroscopy. Furthermore, as solar and galactic cosmic rays can penetrate several meters into Ryugu, these irradiation sources were deemed to be the most likely cause of the differences in the Raman responses observed.

Additionally, MOM/IOM was isolated from a TD1 and a TD2 particle, through demineralization via treatment with HCl and HF [17]. The MOM/IOM was then investigated with FTIR spectroscopy and compared to MOM/IOM from a sample of Orgueil (CI1) run through the entire procedure and then with literature values (Figure 5). While the majority of the band parameters from FTIR spectroscopy indicated that the Ryugu particles were similar to primitive carbonaceous chondrites, including Orgueil, the CH_2_/CH_3_ asymmetric stretching band ratio for the TD1 particle A0035 was anomalously low. The result indicated that the aliphatic material contained within the MOM/IOM consisted of either short chain lengths or highly branched chains. The finding was interpreted as potentially arising from high levels of irradiation [129] at the surface of Ryugu causing the breaking of aliphatic chains and the generation of CH_3_ groups.

Subsequently, another study investigated the MOM/IOM within the Ryugu return samples, also using both Raman and FTIR spectroscopy to study the samples at the µm-scale [130]. In terms of Raman spectroscopy, the study mostly agreed with the previous investigation of Ryugu particles [17], finding that the return samples most closely resembled primitive CI1 and CM2 carbonaceous chondrites. However, while the authors did not interpret their data as showing a difference in irradiation exposure between their TD1 and TD2 samples [130], the TD1 points do plot to lower G-band peak center and higher FWHM values, compared to those for the TD2 particle. Such an observation is similar to that observed by the previous investigation of Ryugu MOM/IOM [17], albeit with fewer data points and a less clear separation of the TD1 and TD2 data.

The FTIR spectroscopic data for the Ryugu samples from the second study [130] were found to also resemble that from primitive CM2 and CI1 carbonaceous chondrites (Figure 5). Nevertheless, a C=O band (~1670 cm^−1^) relating to ketones, aldehydes or amines was observed that has not been detected from any meteoritic MOM/IOM previously. Additionally, the CH_2_/CH_3_ band intensity ratio was found to be 1.9, which is a lot higher than that recorded for the Murchison (1.2) and Ivuna (1.3) carbonaceous chondrites and also both the anomalous value detected for A0035 (0.77) and the carbonaceous chondrite-like value detected for C0008 (1.17) by the first study [17]. The interpretation provided concerning the high CH_2_/CH_3_ is that Ryugu MOM/IOM should contain aliphatic material with longer chain lengths or less branching than that of most primitive carbonaceous chondrites. However, no indication of what might cause this difference was given. One explanation is that the MOM/IOM may have experienced some heating [131], but this is unlikely due to the other data provided concerning Ryugu [17,130]. Alternatively, the MOM/IOM may record an ISM [132,133] or IDP-like [134,135] component (Figure 5), which if true would suggest that Ryugu may have accreted or preserved more primitive ISM/PSN material than carbonaceous chondrites. 

The finding is at odds with that of the first study [17], which indicated that Ryugu MOM/IOM was either similar to carbonaceous chondrites or had a CH_2_/CH_3_ that was lower. While the studies utilized different peak fitting procedures, it may be the case that the Ryugu samples contain a wide diversity of MOM/IOM, which was sourced from distinct environments, including the irradiated asteroid surface, the ISM/PSN and that of the progenitor planetesimal of Ryugu, where aqueous-rich fluids would have been operating.

**Figure 5 life-13-01448-f005:**
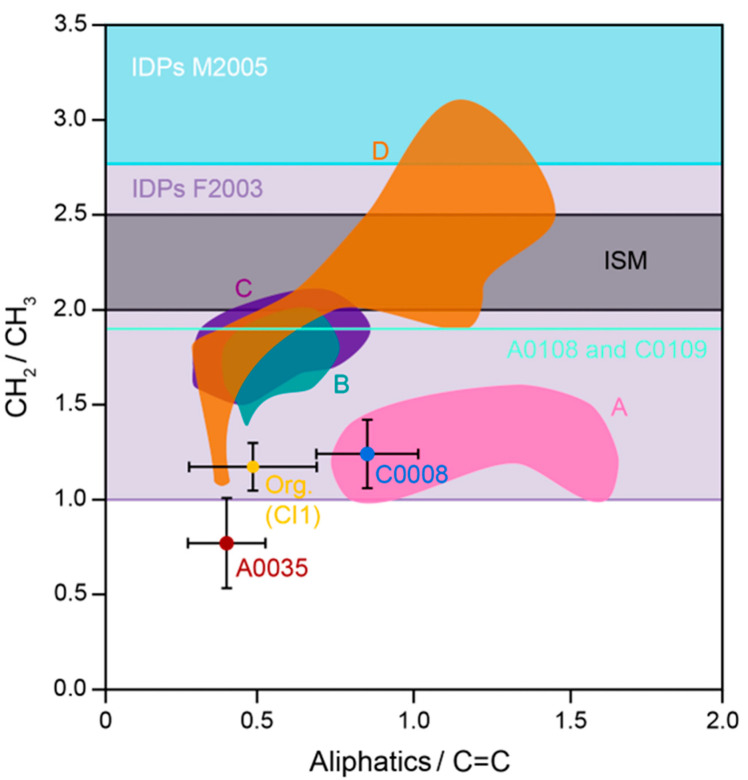
A plot of FTIR band stretching ratios for isolated MOM/IOM from Ryugu [17,130] and carbonaceous chondrites [131] and values measured for IDPs [134,135] and the ISM [132,133]. A = non-heated primitive carbonaceous chondrites, B = slightly heated primitive carbonaceous chondrites and C and D = heated carbonaceous chondrites. It is clear that the CH_2_/CH_3_ ratio for A0035 is anomalously low and those for A0108 and C0109 are anomalously high, compared to that for C0008 and those for primitive carbonaceous chondrites (A and B). Note aliphatics is the sum of the CH, CH2 and CH3 stretching bands. Note that A0108 and C0109 did not have aliphatics or C=C band data available. M2005 = [135] and F2003 = [134].

#### 3.2.2. Nanometer Scale Observations

Transmission electron microscopy (TEM) was employed to understand the OM, petrography and mineralogy of Ryugu particles at the µm to nm scale [17,130]. The results indicated that both µm- and nm-sized MOM/IOM were present throughout the matrix of Ryugu and within carbonaceous nodules [17,130]. Additionally, TEM-EDS and TEM-electron energy loss spectroscopy (EELS) observations were able to distinguish organic nanoglobules and diffuse carbon, which may represent small MOM/IOM films or FOM/SOM [130]. In conjunction with synchrotron-based scanning transmission X-ray microscopy (STXM), X-ray absorption near-edge structure (XANES) and atomic force microscope-based infrared (AFM-IR) spectroscopy, OM classes were defined. The classes were (i) highly aromatic (~25% of OM grains), (ii) aromatic (~35% of OM grains), (iii) a carbonaceous chondrite-like IOM material (~40% of OM grains) and (iv) diffuse carbon, which was often associated with a molecular carbonate peak. While the carbonaceous chondrite-like IOM class describes the OM commonly associated with MOM/IOM in carbonaceous chondrites, it is important to note that this class of MOM/IOM is by no means the only type of MOM/IOM observed in these extraterrestrial samples and the name is therefore somewhat misleading. Furthermore, the frequency of the above classes was found to be consistent between the TD1 and TD2 samples.

For the aromatic class, the STXM response showed that the aromatic-to ketone ratio was higher than for the carbonaceous chondrite-like IOM class. Whereas, for the highly aromatic class a greater diversity of aromatic structures was indicated, due to a broader aromatic peak, compared to the aromatic and carbonaceous chondrite-like IOM classes. Moreover, the particulate and nanoglobular OM was found to be more frequently aromatic or highly aromatic, while the OM dispersed throughout the matrix was mostly present as carbonaceous chondrite-like IOM or diffuse carbon [130]. Additionally, the nanoglobular and particulate MOM/IOM (mostly aromatic or highly aromatic class) was found to be more similar to that found in cometary dust particles or unheated petrographic type 3 carbonaceous chondrites [136,137]. As such, it was interpreted that the nanoglobular and particulate MOM/IOM was most likely to be the originally accreted MOM/IOM and that the functional groups of the other MOM/IOM within the matrix of Ryugu had been influenced by aqueous alteration [130].

When comparing the functional group chemistry of Ryugu to primitive carbonaceous chondrites, it was found that Ryugu contained much higher amounts of the aromatic and highly aromatic MOM/IOM present as nanoglobules and particulate matter [130]. Whereas carbonaceous chondrites possess much more MOM/IOM that is similar to the carbonaceous chondrite-like IOM class. Such a finding is interesting, because it may suggest that Ryugu accreted more primitive MOM/IOM or was at least better able to preserve it, compared to carbonaceous chondrites.

In terms of the diffuse carbon OM, an association with Mg-rich phyllosilicates and molecular carbonate was observed [130]. In conjunction with the functional group observations from MOM/IOM, it was suggested that progressive aqueous alteration leads to the association of the diffuse carbon with phyllosilicates and the introduction of oxygen functional groups into organic particulates and nanoglobules. Additionally, aqueous alteration was also indicated to result in an increase in the variety of XANES spectral features.

#### 3.2.3. Elemental and Isotopic Characteristics

Initial isotopic and elemental measurements of OM in Ryugu samples were performed by elemental analysis-isotope ratio mass spectrometry (EA-IRMS) for bulk sample compositions and secondary ion mass spectrometry (SIMS) for in situ measurements [17]. The results indicated that the majority of the bulk C, H and N element and isotope compositions of the Ryugu particles fell within the range of CM and CI chondrites, but with some plotting to values outside this range. The in situ data indicated the presence of ^15^N and D hot and coldspots for OM, where the hotspots yielded values significantly higher than the bulk values and the coldspots significantly lower. It was found that the ^15^N hotspots or coldspots were the major determiners of the variation in C, H and N isotopes from the bulk values. Furthermore, the deviation in N and H isotopes from the bulk values was found to be both correlated and negatively correlated, with the ^15^N hotspot in the TD1 particle A0073 being correlated with a D enrichment (δ^15^N = 610 ± 78‰, δD = 2983 ± 84‰) and the coldspot in the TD2 particle C0053 being negatively correlated (δ^15^N = −147 ± 10‰, δD = 158 ± 30‰). The results suggested that Ryugu likely accreted OM from various sources, including both ISM and PSN materials and that at least some OM would have been subsequently altered during aqueous alteration, potentially leading to the generation of new MOM/IOM material.

Further investigation of the isotopic and elemental composition of Ryugu return samples was undertaken via EA-IRMS [103] and nano-SIMS [130]. In agreement with the first study [17], the bulk and in situ elemental and isotopic data were found to be mostly within the range of primitive carbonaceous chondrites [103,130]. For MOM/IOM residues recovered after demineralization, the bulk δD values were anomalously low compared to those for isolated MOM/IOM from CM and CI chondrites, which may indicate that the obtained values were influenced by terrestrial water, as a result of either inefficient drying or adsorption of atmospheric moisture onto the OM. For the in situ data, the ^15^N and D coldspots also appeared to be lower than those found previously in carbonaceous chondrites and this was interpreted as relating to the interaction between the OM and the PSN/ISM N_2_ gas. The variation in the D isotopic composition of Ryugu MOM/IOM was interpreted as arising due to varying levels of aqueous alteration which can cause hydrogen isotope exchange and hydrolysis of D-rich MOM/IOM structures. Meanwhile, the variation in the N isotopes was explained by the accretion of OM from different sources. Furthermore, the process responsible for the enrichment in D and ^15^N for hotspots was interpreted as low-temperature fractionation processes within the ISM/PSN [130]. While such processes are the most likely enrichment mechanism for D, they have been determined to be inadequate at fractionating nitrogen isotopes sufficiently to explain the large ^15^N enrichments observed in hotspots within extraterrestrial objects [23,72]. Instead, enrichments in ^15^N are likely the result of photodissociation processes operating in the ISM or PSN [72,87,88,89,90].

When comparing the δ^15^N and δD values for the different studies of Ryugu [17,103,130], it is apparent that all values overlap with those from bulk carbonaceous chondrites [77], IDPs [138], Wild2 cometary particles [82] and comets [79] (Figure 6). While the values for comets, Wild2 particles and IDPs extend to much higher values than those observed for the Ryugu particles, the overlap suggests that at least some of the OM in these extraterrestrial materials bears a resemblance to that in Ryugu.

Further interrogation of the N/C ratio of both the bulk Ryugu particles and HCl treated or MOM/IOM residues from the Ryugu samples revealed differences between the studies [17,103,130]. The first study [17] determined the N and C abundances of both bulk Ryugu particles and HCl-treated aliquots of these, yielding a range of 0.029–0.057 for bulk and 0.032–0.101 for HCl-treated aliquots. The bulk values from the first study overlap with the range of the third study [130] (MOM/IOM residues, 0.021–0.035) (Figure 7), but extend to higher values. For the HCl-treated aliquots of the first study, the values extend up to much higher values than those of the third study (Figure 8). Meanwhile, the second study [103] recorded the bulk values for a single Ryugu particle A0106 and yielded a N/C of 0.043 ± 0.001, which is within the range of the bulk and HCl-treated aliquot values from the first study, but higher than those of the MOM/IOM residues. 

The HCl-treated aliquots from the first study [17] should more closely resemble those of the MOM/IOM residues [130] because the HCl would have removed the carbonates and thus the C abundance measured is the total organic carbon. The discrepancy between the two studies could relate to heterogeneity within the Ryugu samples or the different methods used. In terms of heterogeneity, both studies found that the TD1 samples had higher N/C ratios (0.067 ± 0.021 for HCl aliquots and 0.035 ± 0.006 for the MOM/IOM residues) compared to the TD2 samples (0.043 ± 0.013 for HCl aliquots and 0.021 ± 0.001 for MOM/IOM residues). Accordingly, it would seem that the general N/C relationship between the two touchdown sites is preserved in the values, but the first study is recording higher N/C values compared to the third study. 

As such, it may be that the two techniques used give different values, because of a sample work up or analytical effect. It should be difficult to introduce N-rich material during the HCl treatment of the first study [17], because no nitrogen-rich materials were used during the sample preparation procedure. Furthermore, it is unlikely for the EA-IRMS used in the first study to wrongly record the N/C ratio, because this is the technique commonly used to analyze the stable isotopes of organic matter in bulk meteorite samples and their IOM/MOM residues [15,77,140]. One possible explanation is that some Teflon material is contributing to the bulk signal in the third study [130] and increasing the C abundance. Alternatively, it may be that the third study, which uses nano-SIMS is recording the bulk values of many localized N/C ratios and does not accurately record the bulk range of N/C values.

Nevertheless, all studies found that the N/C ratios overlapped with the range of CI, CM and CR chondrites (Figure 7 and Figure 8) [17,103,130]. However, the HCl-treated aliquot values of the first study [17] ranged to higher values than the other studies, similar to those for IOM/MOM residues from the Tagish Lake meteorite (0.0428 ± 0.0002) and overlapped with the N/C ratio calculated for Wild2 cometary dust particles (0.07–0.24). The range in Ryugu N/C values for all studies also overlapped with the range calculated for particles in the comma of the comet 67P (0.018–0.06) [130,139,140] and the range determined from IDPs (0.02–0.13) [141]. However, the bulk Ryugu N/C values from the first study and N/C values of the second [103] and third studies [130] were lower than those of the dust particles from the comet Wild2 [142]. 

Furthermore, it was suggested that the potential presence of ammonium salts on the comet 67P could significantly raise its N/C ratio if these are taken into account [130,143,144]. Nevertheless, the ammonium salts would consist of small organic molecules and/or sulfate and thus the N/C ratio of any MOM/IOM-like OM in 67P might actually have an N/C ratio similar to the range determined for its coma and overlap with Ryugu. As such, while it is difficult to understand how similar/dissimilar the N/C ratio of Ryugu MOM/IOM is to that in most comets at this time, if 67P, Wild2 and IDPs do probe the same OM [98,99], then the range in Ryugu N/C ratios may overlap significantly with those of comets.

## 4. The Formation and Evolution of Ryugu

### 4.1. Catastrophic Collsion Model

As it currently stands, the most accepted model for the formation and evolution of Ryugu is that of the catastrophic collision and gravitational reaccumulation model (Figure 9) [4,5,145]. In this model, the Ryugu progenitor planetesimal or asteroid was subject to a catastrophic collision, and this created many small debris including boulders and fine particles, which subsequently combined to form Ryugu through gravitational reaccumulation. Indeed, the spinning top-shaped current-day Ryugu has been explained through a combination of a high rotation rate imparted during the collision and the Yarkovsky–O’Keefe–Radzievskii–Paddack (YORP) effect [4,145]. 

A moderately heated and thus dehydrated carbonaceous chondrite-like composition was one of the many attributes assumed by models that aimed to explain the formation of current-day Ryugu [4,5,145]. Such a characteristic was assumed as a result of the remote sensing data that found no band relating to hydrated silicates [5]. It is now known that Ryugu was not dehydrated or heated to temperatures beyond those of primitive carbonaceous chondrites and this is absolutely clear from the organic matter contained within the Ryugu return samples [17,18,103,106,110,130]. Nevertheless, some of the initial collisional models utilized to explain the formation of Ryugu assumed that the collision would be able to explain the heating and subsequent dehydration of Ryugu [5,145]. Whereas other models suggested that the catastrophic collision would not cause significant heating or shock throughout most of the progenitor body [146].

However, the initial input of the collisional simulation required a very large body (~100 km in diameter). It is unclear if the progenitor planetesimal of Ryugu was this large. In fact, such a large progenitor planetesimal may be very unlikely due to previous estimates of the size at which a body would facilitate fluid convection. Bodies greater than ~80 km in diameter would likely facilitate fluid convection and this would ultimately lead to elemental fractionation throughout carbonaceous chondrites [147,148]. Yet no fluid-related elemental fractionation was observed for the majority of Ryugu particles or for carbonaceous chondrites [17,149]. As such, it is unclear whether or not a catastrophic collision would lead to significant heating if the Ryugu progenitor planetesimal was more similar in size to the minimum estimates of ~20 km in diameter, based on the requirements for heat from ^26^Al decay to enable liquid water [17,150].

### 4.2. Cometary Nucleus Model

An alternative to the catastrophic collision and gravitational reaccumulation model is provided by the cometary nucleus model (Figure 10) [10,17,151,152]. In such a scenario, the progenitor planetesimal of Ryugu was broken up by an impact to form ice-rich fragments that would have lost their ice through sublimation to yield rubble pile asteroids. While the planetesimal accretion could have occurred within or near the main belt, it may be more plausible for such an event to have occurred within the outer solar system, due to evidence from Ca and Cr isotopes indicating Ryugu contains the least thermally processed solar system material measured to date [17]. Furthermore, a cometary origin for the Orgueil (CI1) carbonaceous chondrite was proposed previously [102,153] and at current cometary bodies are believed to have formed in the trans-Neptunian disk of the outer solar system [154]. 

Accordingly, the planetesimal progenitor of Ryugu could have formed within the trans-Neptunian disk (TND) and suffered a disruptive event during the massive disk phase, where collisions between trans-Neptunian objects would have been large [155]. The collision could have been either catastrophic [156] or sub-catastrophic [157], and thus the resulting object could have represented a true collisional fragment or a substantially eroded portion of the original body. The subsequent dispersal of the TND that formed the Kuiper belt and Oort cloud [154,158], such as that proposed by the Nice model [159,160,161], could have implanted TND objects into the main belt [162]. Indeed, the main belt has been proposed to have begun empty [163] and active asteroids/main belt comets could represent the descendants of implanted TND objects [154,164,165]. If such a scenario is true, then Ryugu may represent an extinct or dormant comet [10,17,165].

Evidence for past cometary activity on the cometary progenitor of Ryugu was provided by the occurrence of unaltered olivine and pyroxene clasts and the presence of a clast-like domain in the matrix of a Ryugu particle with a distinct lithology [17]. It was suggested that ice sublimation would lead to the generation of gas jets, as seen on the comet 67P and 103P/Hartley 2 [55,166,167]. The jets may have then led to the formation of fractures and caused portions of the cometary surface to collapse. Such events could have trapped any exogenous material picked up by the comet as it traveled through the dusty remnants of collisions within the main belt. The sublimation of ice has also been shown to result in the spin-up of the body and can explain the formation of the spinning top shape of Ryugu [151]. 

Furthermore, cometary jets are thought to deposit a portion of their material back onto the comet’s surface and this can become sintered in place, trapping any organic or dusty material entrained in the jet [55]. Such a process was employed to explain some of the geochemical differences observed between TD1 and TD2 particles and the larger geochemical heterogeneity found among the TD1 particles [17]. As material from many different depths can be entrained within the jets, if jets were operating in the past at or near the TD1 site, material from distinct regions of the cometary progenitor of Ryugu could be accumulated within a narrow region on its surface. Meanwhile, if the TD2 site did not experience jet activity the samples may just record a single region of Ryugu and thus appear more similar.

In terms of organic matter, Ryugu appears to contain CI-chondrite-like organic matter including its amino acids. Accordingly, it has been proposed that CI chondrites, such as Orgueil, may have a cometary origin due to the distinct differences between their amino acids and those of CM2 chondrites [168]. Furthermore, the discovery that Ryugu contains more aromatic or highly aromatic material related to organic particulates or nanoglobules [130] than carbonaceous chondrites may imply that it accreted a more primitive composition. In agreement, the finding that Ryugu MOM/IOM has a high CH_2_/CH_3_, which is similar to that observed in IDPs and for ISM material, may indicate that Ryugu accreted more comet-like or ISM-like organic matter than most carbonaceous chondrites. Together with the overlap of H and N isotopic values and N/C ratios for Ryugu organic matter and values determined for some comets, the notion of a cometary origin for Ryugu is certainly plausible. Moreover, a cometary origin for Ryugu proposes the advantage of not requiring a catastrophic impact of a rocky asteroid, which may cause significant heating and shock-related features within the resulting material, both of which are not observed In the Ryugu return samples.

## 5. Conclusions

The Hayabusa2 mission to the asteroid Ryugu was the first mission to return samples from a primitive C-complex asteroid to Earth and has been a great success, with all of the primary scientific objectives having been completed. The array of techniques that have been applied to such a small sample size is truly exceptional. The Ryugu return samples contain abundant OM, which is similar in nature to that observed within primitive carbonaceous chondrites (especially CI1 chondrites), but also shows some potential similarity to that thought to be contained within comets or IDPs. Among the most important findings is the discovery that Ryugu experienced heterogenous aqueous alteration and that this affected its organic materials, including some of the building blocks of life. Furthermore, the possibility that Ryugu amino acids are racemic, questions the role of astrophysical environments in the determination of single enantiomers being utilized by life on earth. 

Nevertheless, while a number of studies have now reported their findings concerning the OM contained within Ryugu, the limited sample size and associated challenges of carrying out analyses on such samples mean there are still many unknowns or ambiguities that need to be elucidated. For example, the reason why Ryugu shares a more similar N-heterocycle (including nucleobase) inventory to Murchison, compared to CI1 chondrites, the nature of the abundances and enantiomeric excesses of Ryugu-hydrolyzed amino acid extracts and the reason for the distinct functional group chemistry of Ryugu MOM/IOM remain to be resolved. Moreover, it still remains unclear whether irradiation has modified the surface OM inventory of Ryugu, or if planetesimal processes, such as aqueous alteration, or accretion of OM from heterogenous sources, are responsible for the variation observed between Ryugu particles in terms of their Raman responses and FOM/SOM contents. Additionally, the large variation in Ryugu MOM/IOM CH_2_/CH_3_ ratios need further investigation to better understand what processes and organic matter sources are responsible and if Ryugu is truly comparable in terms of its OM to carbonaceous chondrites observed so far. Therefore, in order to obtain the most from the Hayabusa2 sample return mission, future work should focus on the differences between different Ryugu particles and the ever-growing number of extraterrestrial samples contained within collections on Earth.

In terms of the formation and evolution of Ryugu, both the catastrophic collision model and the cometary nucleus model provide potentially viable explanations for the origin of Ryugu, but further work is required to better constrain which model is most likely. Such future work should aim to achieve a better understanding of cometary OM and internal structure and include more advanced modeling of both the catastrophic disruption of planetesimals and the dynamic evolution of comets.

## Figures and Tables

**Figure 1 life-13-01448-f001:**
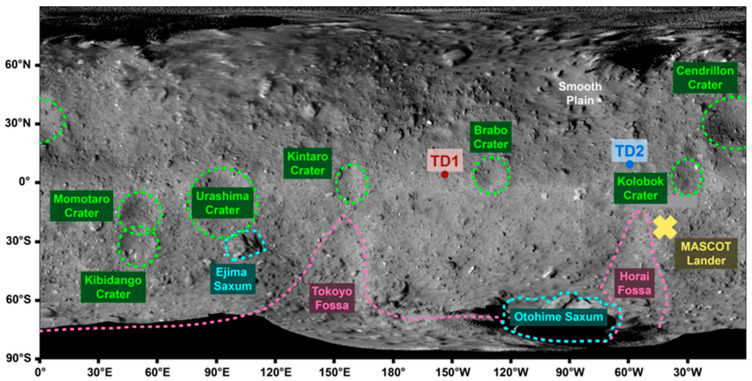
A map of Ryugu showing the key geological features and the touchdown sites from which the samples were located. TD1 = touchdown site 1 and TD2 = touchdown site 2. Credit for the background image belongs to the Japan Aerospace Exploration Agency (JAXA), University of Tokyo, Kochi University, Rikkyo University, Nagoya University, Chiba Institute of Technology, Meiji University, University of Aizu and National Institute of Advanced Industrial Science and Technology AIST. The key geological features were overlayed onto the map by consulting with the literature [11].

**Figure 2 life-13-01448-f002:**
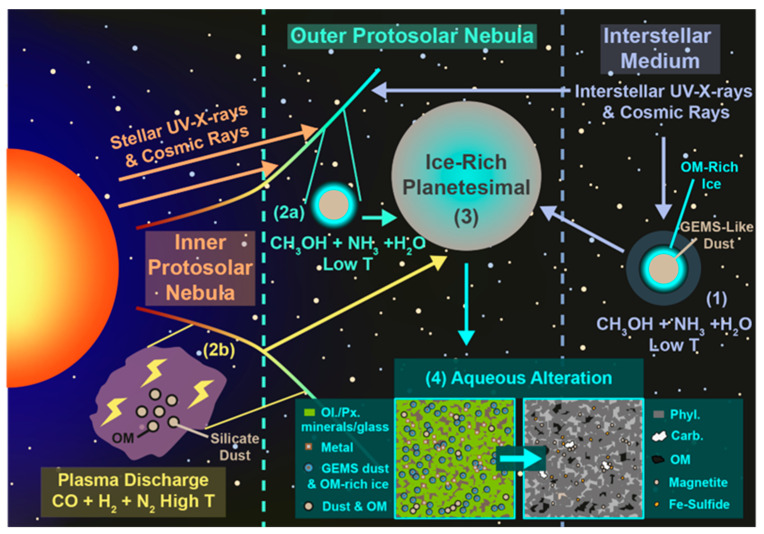
A diagram outlining the main formation environments for organic matter (OM) found within the ice-rich progenitor planetesimals of meteorites, asteroids and comets. Note that this figure was adapted from Figure 3 in a previous study [18], more details concerning the license under which the work was published can be found at: https://creativecommons.org/licenses/by/4.0/, (accessed on 30 May 2023). (1) OM can form in the interstellar medium (ISM) within molecular clouds at low temperatures (Low T) prior to the formation of the Solar System. (2) OM can also be formed within the protosolar nebular (PSN) or protoplanetary disk (PPD) at low temperatures (2a) or high temperatures (High T) at the surface of the PSN exposed to the proto-Sun (2b). (3) OM formed prior to or during the accretion of planetesimals can become accreted into such bodies. (4) Short-lived radionuclides, e.g., ^26^Al, decay and generate heat, which melts organic-bearing ice that leads to aqueous alteration and the alteration/generation of organic matter and minerals. Ol. = olivine or similar composition amorphous silicates, Px. = pyroxene or similar composition amorphous silicates, GEMS = glass with embedded metal and sulfides, Phyl. = phyllosilicates, and Carb. = carbonate.

**Figure 6 life-13-01448-f006:**
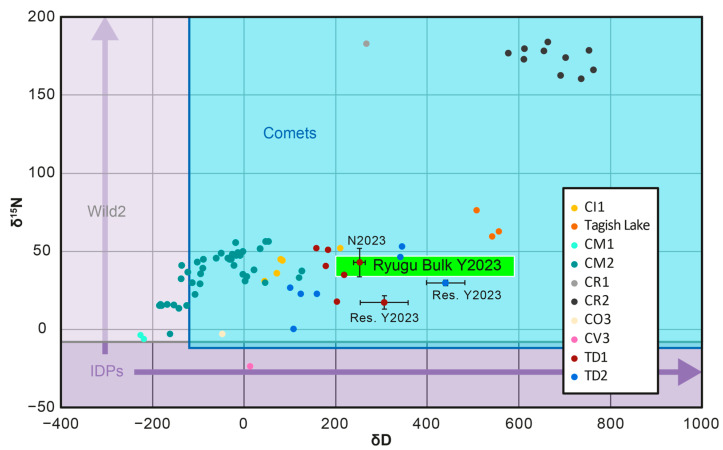
A plot of δ^15^N versus δD for bulk Ryugu particles from TD1 and TD2 [17,103], isolated Ryugu MOM/IOM residues [130], bulk carbonaceous chondrites [77], IDPs [138], Wild2 cometary particles [82] and comets [79]. Note that N2023 = [103], Y2023 = [130] Res. = residue and that the purple arrows indicate the range of the values for IDPs. It is clear that the values for Ryugu, CM2, CI1, IDPs, Wild2 particles and comets overlap. Meanwhile Tagish Lake and CR chondrites plot to higher δ^15^N versus δD values.

**Figure 7 life-13-01448-f007:**
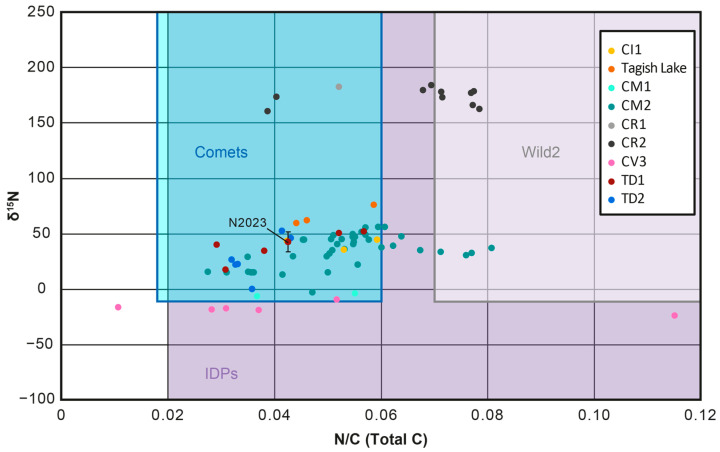
A plot of the δ^15^N versus the N/C (total carbon) ratio for bulk Ryugu particles [17,103] and carbonaceous chondrites [77]. The δ^15^N and N/C Values for IDPs, Wild2 cometary particles and comets are included for comparison. It is clear that the values for Ryugu particles overlap with those for CM2 and CI1 chondrites, the Tagish Lake chondrite, IDPs and comets, but fall to lower δ^15^N values compared to CR chondrites and lower N/C ratios compared to Wild2 particles. Note that total carbon refers to measurements of the bulk particles or meteorites, without isolation of the MOM/IOM or removal of other carbon bearing phases such as carbonates. For comets, the range in δ^15^N includes the values from many cometary bodies, but due to the lack of N/C information for the majority of these bodies the N/C ratio given is the range determined for 67P [80,139].

**Figure 8 life-13-01448-f008:**
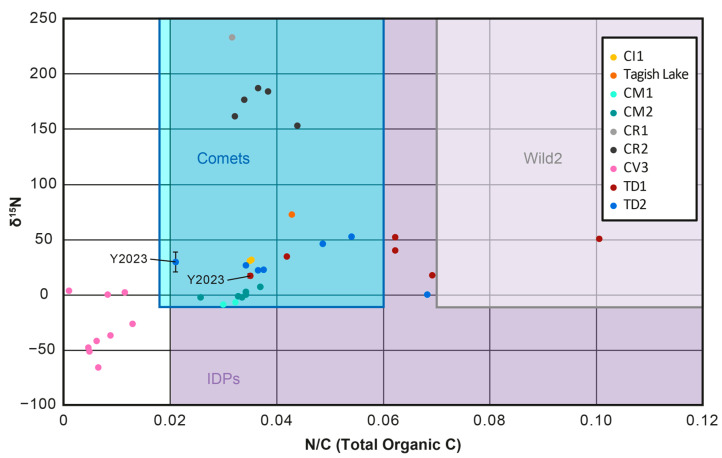
A plot of the δ^15^N versus the N/C (total organic carbon) ratio for HCl treated Ryugu particles [17] or MOM/IOM isolated from them [17,103] and MOM/IOM isolated from carbonaceous chondrites [77]. The δ^15^N and N/C Values for IDPs, Wild2 cometary particles and comets are included for comparison. It is clear that the values for Ryugu particles overlap with those for CM2 and CI1 chondrites, the Tagish Lake chondrite, IDPs, comets and Wild2 particles, but fall to lower δ^15^N values compared to CR chondrites. Note that total organic carbon refers to measurements of Ryugu samples, where the MOM/IOM has been isolated or major C-bearing phases other than OM have been removed by HCl (e.g., carbonate). For comets, the range in δ^15^N includes the values from many cometary bodies, but due to the lack of N/C information for the majority of these bodies the N/C ratio given is the range determined for 67P [80,139].

**Figure 9 life-13-01448-f009:**
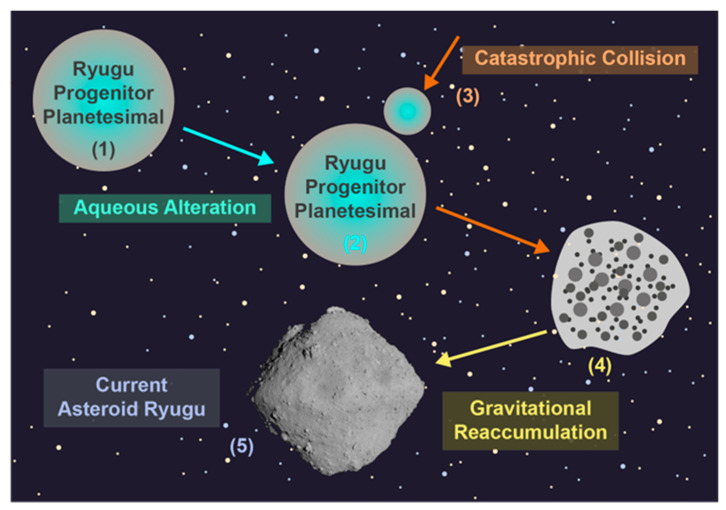
An illustration of the catastrophic collision and gravitation reaccumulation model. (1) The Ryugu progenitor planetesimal forms via the accretion of silicate dust and water-rich ice. (2) aqueous alteration leads to the alteration of the silicate dust and the formation of new minerals. Organic matter accreted within the ice is also altered to form new OM. (3) an impactor collides with the planetesimal reducing the body to debris. (4) the debris reaccumulates due to gravity and forms a rubble pile asteroid. (5) Rotation imparted by the impact and from the YORP effect leads to the generation of a spinning top-shaped asteroid and its movement into a near-Earth orbit. Credit for the image of Ryugu belongs with JAXA.

**Figure 10 life-13-01448-f010:**
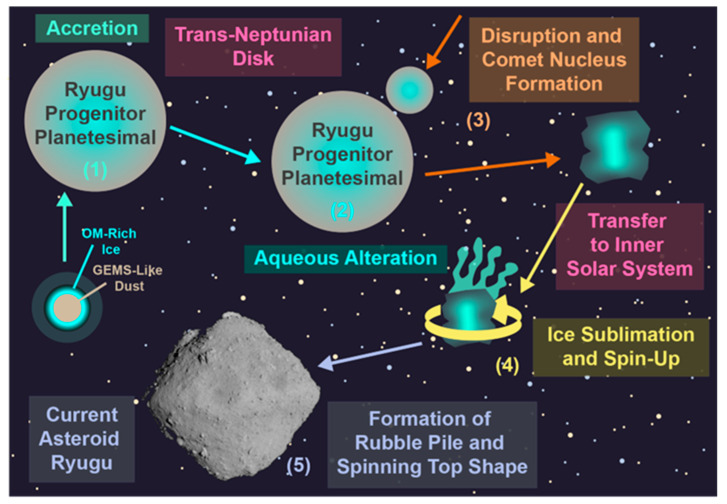
An illustration of the cometary nucleus model. (1) The Ryugu progenitor planetesimal forms via the accretion of silicate dust and water-rich ice within the trans-Neptunian disk (TND). (2) aqueous alteration leads to the alteration of the silicate dust and the formation of new minerals. Organic matter accreted within the ice is also altered to form new OM. (3) during the massive disk phase of the TND, an impactor collides with the planetesimal causing the body to break up into many large chunks (catastrophic impact) or multiple impact events lead to the erosion of the planetesimal to leave behind a much smaller body (sub-catastrophic impacts). (4) After the dispersal of the TND, via a Nice model-like scenario, the Ryugu progenitor planetesimal fragment is transferred into the inner Solar System. In the inner Solar system, the fragment displays cometary behavior, due to the loss of volatiles, which in turn spin-up the body. (5) as a result of the spin-up, the rotation speed of the body increases, and this forms a spinning top shape for the material left behind. Eventually, the body sublimes off all of its surface ice and a rubble pile asteroid is formed. The current day Ryugu may thus resemble an extinct or dormant cometary nucleus. Credit for the image of Ryugu belongs with JAXA.

## Data Availability

All the data presented in this review can be found within the cited manuscripts.

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
