# Peer review of "Organic Matter in the Asteroid Ryugu: What We Know So Far"

_life, 2023, doi:10.3390/life13071448_

Round 1

Reviewer 1 Report

This is an excellent paper.  My only significant comment relates to their conclusions.  I believe the authors need to go beyond their statement that "further work is required to better constrain which model is most likely."  I believe they should specify more specifics and perhaps discuss what other asteroid samples might help constrain the origin models.

Reviewer 2 Report

An exhaustive review dedicated to the important results of the Hayabusa 2 mission. Of particular interest is the discovered complex organic matter, especially optically active. But how could it be safely delivered to the planets of the Earth group? This issue requires separate consideration and is beyond the scope of this article.

Reviewer 3 Report

The review paper on the organics found in the samples of the asteroid Ryugu presents a very nice summary of the findings. This is a valuable contribution for all scientists dealing with organics, be it in meteorites, asteroids or comets.

I have only some minor comments, which are annotated to the manuscript.
